# Effect of Gliding Arc Plasma Jet on the Mycobiota and Deoxynivalenol Levels in Naturally Contaminated Barley Grains

**DOI:** 10.3390/ijerph20065072

**Published:** 2023-03-14

**Authors:** William Chiappim, Vanessa de Paula Bernardes, Naara Aparecida Almeida, Viviane Lopes Pereira, Adriana Pavesi Arisseto Bragotto, Maristela Barnes Rodrigues Cerqueira, Eliana Badiale Furlong, Rodrigo Pessoa, Liliana Oliveira Rocha

**Affiliations:** 1Laboratory of Plasmas and Applications, Department of Physics, Faculty of Engineering and Sciences, São Paulo State University (UNESP), Guaratinguetá 12516-410, Brazil; 2Laboratório de Microbiologia de Alimentos I, Departmento de Alimentos e Nutrição, Faculdade de Engenharia de Alimentos, Universidade Estadual de Campinas-UNICAMP, Campinas 13083-862, Brazil; 3Escola de Química e Alimentos, Universidade Federal do Rio Grande, Rio Grande 96203-900, Brazil; 4Laboratório de Plasmas e Processos, Departamento de Física, Instituto Tecnológico de Aeronáutica, São José dos Campos 12228-900, Brazil

**Keywords:** cold plasma, decontamination, mycotoxins, *Fusarium graminearum*, *Fusarium meridionale*, QuEChERS, masked mycotoxins

## Abstract

*Fusarium graminearum* and *Fusarium meridionale* are primary contaminants of barley, capable of producing several mycotoxins, mainly type B trichothecenes and zearalenone. Cold plasma decontamination has been gaining prominence, seeking to control the fungal and mycotoxin contamination of food and feed and to improve product quality. To reach this objective, the present study was divided into two parts. In the first part, *F. meridionale* and *F. graminearum* strains were exposed to gliding arc plasma jet (GAPJ). Cell viability tests showed the inactivation of *F. meridionale* after 15-min treatment, whereas *F. graminearum* showed to be resistant. In the second part, barley grains were treated by GAPJ for 10, 20, and 30 min, demonstrating a reduction of about 2 log CFU/g of the barley’s mycobiota, composed of yeasts, strains belonging to the *F. graminearum* species complex, *Alternaria*, and *Aspergillus*. A decrease in DON levels (up to 89%) was observed after exposure for 20 min. However, an increase in the toxin Deoxynivalenol-3-glucoside (D3G) was observed in barley grains, indicating a conversion of DON to D3G.

## 1. Introduction

The exposure of non-thermal plasma (cold plasma) to the environment generates the so-called non-thermal atmospheric pressure plasmas (NTAPPs) [1]. NTAPP is an eco-friendly technology generated by an ionized gas that produces reactive species from the gas phase of plasma, plasma–air interaction, or interaction with treated/activated substrates [2]. Reactive species are of great scientific interest due to the action of their active constituents in areas such as, medicine, dentistry, agriculture, environment, and food [1,3,4,5,6,7,8,9]. The physicochemical reactions generated in plasma interaction with air create reactive species of oxygen and nitrogen (RONS), whose main constituents are nitrates (NO_3_), nitrite (NO_2_), peroxynitrite (ONOO^−^), hydrogen peroxide (H_2_O_2_), hydroxyl (OH), oxygen (O_2_), superoxide (O_2_), and ozone (O_3_) [2,3,5,10]. These plasmas are generally generated as plasma jets (PJ) and may be applied directly or indirectly [1,8,11,12,13]. Direct application is when the PJ is used directly on a substrate for surface treatment or activation. In contrast, indirect application is when plasma activates a liquid that will be used for a specific application [2,14]. The most used discharges for PJ generation are dielectric barrier discharge (DBD) and gliding arc (GA) [11,12,15,16,17,18,19,20,21]. However, the nature of each unloading application may vary slightly within the same area of knowledge; for example, the DBD plasma jet is best suited for wound and tumor treatments [22,23,24,25] due to lower operating temperature, and does not generate an arc, which can cause a burn on living tissues [7]. In contrast, for the activation of large volumes of liquids and in the inactivation of fungi, the PJs that generate higher concentrations of RONS are the GA [12,19,20].

Barley is one of the most produced cereals worldwide, occupying the fourth position of economic importance [26]. However, the growth of filamentous fungi in barley is a concern for the food industry and it mainly happens due to inappropriate management during the pre-harvest, transportation, and storage of cereal grains [27]. In addition, climatic conditions associated with insect infestation and intrinsic factors of the raw material are considered relevant, as they may favor the development of these microorganisms and mycotoxin production, which are toxic secondary metabolites mainly produced by the genera *Alternaria*, *Aspergillus*, *Fusarium*, and *Penicillium*, with the *Fusarium graminearum* species complex (FGSC) often reported in barley (dominant species, with more than 80% frequency) [28,29,30]. Another species that is important to study is *F. meridionale*, which significantly contributes to the Fusarium Head Blight (FHB) disease of wheat in Paraná and Rio Grande do Sul, Brazil. Mainly due to the crop rotation of wheat, corn, and barley in these states, there is evidence of barley grain contamination by FGSC strains from wheat and corn, as recently demonstrated by Machado et al. [31] in the cultivation of barley in the state of Rio Grande do Sul, Brazil.

Mycotoxins are low molecular weight compounds that can cause toxic effects to vertebrate animals, even when consumed at low concentrations [32]. Among the most severe impacts described are carcinogenicity and hepatotoxicity [32]. In addition, mycotoxins are chemically stable and tend to be found in food commodities throughout the food chain; therefore, monitoring these compounds from field production to final processing by food industries and government agencies are of utmost importance [33].

Among the mycotoxins most found in barley grains, deoxynivalenol (DON), other type B trichothecenes (nivalenol, 3- and 15-acetyldeoxynivalenol) and zearalenone (ZEN) stand out, which are produced mainly by the FGSC [29,34]. *Fusarium graminearum sensu stricto* (*s.s*.) is the predominant species in small-grain cereals, including barley, and negatively affect the plant’s health in the field, causing the disease FHB, which causes damage to the grains, leading to productivity loss, decreased germination capacity, and reduced malt quality [35,36]. In addition, grains affected by the disease have high levels of type B trichothecenes, especially DON, causing rejection by the food and beverage industry and economic loss [37].

In humans and animals, DON can cause nausea, vomiting, and immune system suppression. These toxic effects are caused due to the inhibition of protein and nucleic acid synthesis by DON at the cellular level. In plants, this toxin inhibits both germination and growth; it is important to highlight that the plant host can metabolize DON into deoxynivalenol-3-glucoside (D3G), a “masked mycotoxin” with reduced toxicity towards plants [38]. D3G has been widely reported in wheat and barley as the main metabolite of DON; this compound can be hydrolyzed into its parent mycotoxin through the intestinal mycoflora of animals, contributing to the overall risk of DON exposure. Studies have also shown that D3G can be formed after food processing, such as fermentation, baking, and brewing [39,40].

To eliminate contamination by fungi and/or mycotoxins from food commodities and improve product quality, several decontamination technologies have been developed, such as gamma radiation [41], ozonation [42], enzymatic treatments [43], and the use of mycotoxin adsorbents in animal feed [44]. In this context, recently, PJ has shown promising results to control the occurrence of fungi and mycotoxins in food [45]; the reactive species present in NTAPP may inactivate microorganisms, leading to increased product safety and prolonged shelf life [46,47,48].

In filamentous fungi, NTAPP can act on the cell wall and membrane, resulting in loss of structure. In addition, it can cause oxidative stress of the cell, causing changes in nucleic acids and, therefore, cell death [47]. Regarding mycotoxins, studies have shown that NTAPP provides stability loss, resulting in degradation products that may have lower toxicity than the original molecule [47,49].

Due to its antimicrobial properties, PJ is considered a technology of contemporary interest for the food industry [47]. The rapid decontamination of food under conditions of room temperature and atmospheric pressure without incurring excessive costs when using air as the working gas makes PJ particularly attractive to the market. Some of the food sectors that could benefit from PJ’s antimicrobial property include fresh produce, grains and oilseeds, spices, dried meats, dairy products and derivatives, and the fish industries [50].

Considering that barley is one of the important cereals for the economy and its high potential for FGSC contamination and its mycotoxins, this work aims to evaluate the effects of gliding arc plasma jet (GAPJ) on the FGSC (*F. graminearum s.s.* and *F. meridionale*) and on the mycotoxin DON in naturally contaminated barley grain samples.

## 2. Materials and Methods

### 2.1. Strains of the Fusarium graminearum Species Complex

Barley grain samples were obtained from the December 2017 harvest in the State of Rio Grande do Sul, Brazil, and were kindly provided by Dr. Euclides Minella from the Brazilian Agricultural Research Corporation (EMBRAPA). The samples underwent a cleaning and drying process at 60 °C and were subsequently stored in 1.0 kg polyethylene bags at 4 °C.

*F. meridionale* (FML03) and *F. graminearum* (FML08), isolated from barley grains from Rio Grande do Sul (RS), Brazil, were previously characterized by Iwase et al., 2020 [51], based on *RPB1* and *RPB2* loci. The sequences were deposited into the NCBI (National Center for Biotechnological Information) database under the accession numbers MT118772—*RPB1* and MT119305—*RPB2* for FML03; MT118786—*RPB1* and MT119317—*RPB2* for FML08. These species were selected due to their high occurrence in Brazilian barley grain samples [51,52,53,54]. Each strain was grown on five Petri dishes (150 × 20 mm) containing potato dextrose agar (PDA) under continuous light for fifteen days at 25 °C. After this period, the colony’s surface formed was lightly scraped with a sterile scalpel, and the aerial mycelium was transferred into a tube containing 20 mL of sterile distilled water. The spore count was performed in a Neubauer chamber, and the final suspension was adjusted to 1 × 10^8^ spores/mL. Subsequently, the spore suspension was transferred to a PDA medium for growth for ten days. Fragments of an area of 0.70 cm^2^ were removed with a scalpel from 2.5 cm from the central region of the culture grown in PDA.

#### 2.1.1. Effect of Gliding Arc Plasma Jet on *F. graminearum* and *F. meridionale*

The effect of GAPJ on fungal growth was evaluated according to the methodology proposed by Phan et al., 2019 [55], with modifications. After fungal growth, as described in the previous item, the area fragments (0.7 cm^2^) were transferred to the center of a Petri dish containing PDA. The following samples were evaluated: (a) positive control (untreated fragment); (b) negative control (no fungal contamination); (c) treatments with GAPJ with exposure for 5, 10, and 15 min. The experiments were performed in triplicate. Subsequently, the radial growth was analyzed. For that, the plates were incubated at 25 °C for 15 days, and the radial growth rate was evaluated according to the method described by Patriarca et al., 2001 [56].

Figure 1 shows the configuration used to study the inactivation of FML03 and FML08. The electrical parameters of the GAPJ device were characterized according to Chiappim et al., 2021 [14]. Table 1 shows the main parameters of the plasma jet and the substrate. It is important to emphasize that the plasma did not come into direct contact with the PDA. In the process, reactive oxygen and nitrogen species interact with the PDA medium.

The electrical discharge device used is a GAPJ generated in a forward vortex flow reactor (FVFR) with compressed air (Schulz CSD 9/50, Joinville, SC, Brazil). The FVFR produces a gliding arc plasma jet (GAPJ) between the spark plug that is the cathode and an anode made of stainless steel and designed in a cylindrical shape. The experimental setup comprises a plasma reactor, a high voltage power supply (model Arternis 0215, Inergiae, Florianópolis, SC, Brazil), an oscilloscope (Keysight DSOX1202A, Keysight, Santa Rosa, CA, USA), a probe voltage (Tektronix P6015A, Tektronix, Beaverton, OR, USA), and a self-adjusting current probe (Agilent N2869B, Agilent, Santa Clara, CA, USA).

#### 2.1.2. Effect of Gliding Arc Plasma Jet on Cell Viability of *F. graminearum* and *F. meridionale*

To investigate the potential of GAPJ treatment to induce cell death in FML03 and FML08, cell viability indicators, namely thiazole orange (TO) and propidium iodide (PI), were used, according to the methodology described by Braschi et al., 2018 and Rocha et al., 2013 [57,58]. It is worth mentioning that the TO dye is permeable to both viable and non-viable cells, whereas PI is permeable only to non-viable cells. The exposure time to GAPJ of 15 min was selected for this analysis due to better results on the fungicidal effect and fungistatic effect on *F. meridionale* and *F. graminearum*, respectively. The procedure was conducted as followed: after a 15 min exposure to GAPJ, the fungal mycelium was removed from the culture medium and transferred to a 1.5 mL microtube. Subsequently, 300 µL of phosphate-buffered saline (PBS) was added. The suspension was shaken and centrifuged at 12,000 rpm for 15 min and washed. Then, 50 µL was transferred to another microtube, and the dyes were added (3 µL of TO at 42 µmol/L and 1 µL of PI at 4.3 mmol/L). The aliquots were transferred to slides and visualized under a confocal microscope (Leica TCS SP5 II, Leica, Wetzlar, Germany). For TO, the excitation and emission wavelengths were 514 and 525–564 nm, respectively; whereas for PI, the excitation and emission were 543 and 608–704 nm, respectively.

### 2.2. Barley Grains Treated by Gliding Arc Plasma Jet

Samples of barley grains (20 g) naturally contaminated by the FGSC and DON from Rio Grande do Sul state, Brazil, were exposed to GAPJ to evaluate the potential for reducing fungal contamination. For this, the experimental setup shown in Figure 2 was used. The barley grains were placed inside a glass cylinder with a nozzle to transport the reactive species generated in the non-thermal plasma. A magnetic stirrer (Kasvi CK FG, Kasvi, São José dos Pinhais, PR, Brazil) was used at 3000 rpm to homogenize the treatment of the grains. The GAPJ nozzle was positioned 3 cm from the grain surface. The treatment times were 10, 20, and 30 min. The experimental setup used for the electrical characterization of the plasma is the same as the one described in the previous section. The treated and untreated samples (control) were analyzed in quintuplicate; in total, 20 samples were analyzed.

It is essential to highlight that to evaluate the negative effect of RONS on barley grains exposed to plasma for extended periods, the following indicators need to be used: (i) decrease in chlorophyll content; (ii) increase in lipid peroxidation; (iii) reduction in photosynthetic efficiency; (iv) oxidative damage to DNA and proteins; and (v) alteration in antioxidant enzyme activity. These indicators can be measured using various techniques such as spectrophotometry, high-performance liquid chromatography (HPLC), and enzyme-linked immunosorbent assay (ELISA). By monitoring these indicators, it is possible to determine the extent of damage caused by RONS to barley grains and to develop strategies to mitigate their harmful effects. Although, this was not the focus of the present research, future investigation on this matter should be considered.

Figure 3 shows another point of paramount importance in validating the present research: the temperature as a function of the time of exposure to the plasma. As can be seen, the maximum treatment temperature is 38 °C for both cases. It is known that the temperature treatment required to eliminate *F. graminearum* and *F. meridionale* may depend on factors such as specific growing conditions, growth phase, and type of material to be sterilized. Generally, standard sterilization methods include autoclaving and dry heat (>100 °C). The appropriate temperature and duration for each process may vary depending on the specific conditions and material to be treated. Barley grains used in this study were previously submitted to drying at 60 °C, and this was above the maximum temperature reached by the GAPJ treatment. Additionally, even after this relatively high drying temperature, *F. graminearum* and *F. meridionale* were recovered from the samples. Therefore, we credit any reduction in fungal contamination to the plasma treatment.

#### 2.2.1. Mycobiota of Barley Grains

For the isolation of fungi from the control and treated samples, PDA medium was used [59]. Approximately 10 g of barley grains was taken from each sample, ground, and transferred to 90 mL of sterile peptone water. After stirring for three minutes, 1 mL was transferred to 9 mL of sterile distilled water, and successive dilutions were carried out to a dilution of 0.001. Then, an aliquot of 0.1 mL of each dilution was transferred to Petri dishes containing PDA medium, and the analysis was carried out in triplicate [59]. The Petri dishes were incubated for seven days at 25 °C, and the counts were performed in log CFU/mL. Colonies belonging to different morphological types were classified according to genus level. The isolates belonging to *Fusarium* genus were identified based on morphological characters on carnation leaf agar (CLA), and then, the FGSC strains were counted in log CFU/mL.

#### 2.2.2. Analysis of Deoxynivalenol and Deoxynivalenol-3 Glucoside

For DON and D3G analyses, the 20-min GAPJ treatment was chosen because it was considered adequate to reduce the FGSC counts in barley. Twenty barley samples from RS, Brazil, were used for this experiment, and mycotoxin analyses were conducted in non-treated (control) and GAPJ-treated grain samples.

To extract DON and D3G mycotoxins, the Quick, Easy, Cheap, Effective, Rugged, and Safe (QuEChERS) method was used. Five grams of barley grains were ground and homogenized with 10 mL of formic acid (0.1%) + 10 mL of acetonitrile. Then, the samples were shaken for 20 min at 250 rpm. After this process, 4 g of MgSO_4_ + 1 g of NaCl were added, and the samples were centrifuged at 5000 rpm for 5 min. Finally, 0.5 mL of the mixture was withdrawn and diluted in an equal volume of deionized water. Samples were filtered through a 0.2 µm nylon microfilter [51].

The chromatographic analyses were based on the method described by Arraché et al. 2018 [60]. DON and D3G were determined by a high-performance liquid chromatograph (HPLC) system (Nexera-UC, Shimadzu, Kyoto, Japan) consisting of a SIL-20AHT autoinjector (volume of 20 µL) and a GEMINI 5 µm C18 chromatographic column (4.6 × 250 mm), and the mobile phase was composed of phase ultrapure water: acetonitrile (30:70) with isocratic mode and a flow rate of 0.5 mL/min. A diode array detector was used, with a maximum absorption wavelength of 220 nm [60].

Data were collected through LC-solution Shimadzu software. The limit of detection (LOD, signal to noise ratio = 3), limit of quantification (LOQ, signal to noise ratio = 10), and recovery were 15 µg/Kg and 62.5 µg/Kg and 90%, respectively, for both mycotoxins [48].

### 2.3. Statistical Analysis of Data

Variant analysis (ANOVA) was used for statistical analysis. The Tukey test was used to verify the results on the mycobiota and mycotoxin contamination. A *p*-value < 0.05 was considered statistically significant. Pearson’s correlation test assessed the relationship between GAPJ treatment time and colony counts.

## 3. Results and Discussion

### 3.1. Effect of Gliding Arc Plasma Jet on F. graminearum and F. meridionale Strains

Figure 4 shows the radial mycelia growth of FML03 and FML08 during 3, 5, 10, and 15 days at 25 °C after GAPJ exposure times of 5, 10, and 15 min. Measurements were performed daily to understand the inhibition mechanism of *F. meridionale* (FML03) and *F. graminearum* (FML08). As can be seen, a fungistatic effect was obtained in the species *F. meridionale* during the first three days when exposed to GAPJ for 5 and 10 min. However, after this period, fungal growth was observed until the 15th day.

In contrast, after 15 min of exposure, a fungicidal effect was observed in FML03, demonstrated by an absence of growth for 15 days. Recent studies have reported that prolonged exposure of microorganisms to RONS produced by GAPJ has antimicrobial effects, proving the effectiveness of plasma treatment [61]. In addition to RONS, fungi under plasma treatment undergo intense reactions with electrons and ions, and, for this reason, membranes and cell wall components suffer damage [62]. On the other hand, UV radiation is responsible for the breakage of DNA strands, which results in mutations that corroborate with the fungicidal or fungistatic effects [63]. As seen in Figure 4, *F. graminearum* (FML08) was more resistant to treatment with GAPJ, which is demonstrated by the fungistatic effect at all times of exposure to GAPJ. For this reason, an additional test was performed for 20 min, and the growth of FML08 was completely inhibited for 15 days (data not shown). Differences in the susceptibility of *F. meridionale* and *F. graminearum* to GAPJ can be correlated with cellular components, such as the fungal cell wall, which has several peculiarities and differences from species to species [64].

In agreement with our previous work [14], the optical emission spectrum of the GAPJ contains NH, N_2_, and OH species in abundance. In addition to these species, the literature shows that GAPJ interacting with air generates the so-called peroxynitrite (NO, NO_2_, NO_3_^−^, OH, H_2_O_2_, O_3_, O, ^1^O_2_, HO_2_, O_2_*, HNO_2_) [2]. It is important to note that the ozone concentration generated by the GAPJ is relatively low when operated in ambient air. As reported by Pawlat et al. [65], a small amount of ozone is produced due to the oxidation reaction of NO_x_ with long-lived oxygen species, including ozone; this behavior is due to the temperatures reached by the GAPJ, exceeding 35 °C [66]. Therefore, this behavior highlights the unique chemical environment of the GAPJ and its potential to generate a variety of reactive species with diverse applications.

Previous studies suggest that the acidification of the water is due to the presence of peroxynitrites that act synergistically, generating an acidic environment that is conducive to damaging biological molecules, such as membranes and DNA [67,68,69].

In general, these results demonstrate the effectiveness of GAPJ treatment in inhibiting fungal growth, which may be an alternative for controlling the spoilage of fruits, vegetables, and cereals after harvest.

Although we did not find specific information on cell wall components of *F. graminearum* and *F. meridionale*, their differences could partially explain the distinct response to GAPJ treatment. It is worth noting that the cell wall composition of filamentous fungi is complex and may vary depending on many factors, including genetic variation among strains, the fungal growth stage, their associated host, and specific environmental conditions. *F. meridionale* and *F. graminearum* are closely related species; nevertheless, their phenotypic differences are significant and involve variation in mycotoxin production, adaptation and aggressiveness towards plant hosts, and others [70]. Therefore, it was expected a distinct response to plasma treatment. The reasons for these differences are hard to determine, however studies involving strain metabolic response to stress and specific morphological characteristics (i.e. cell wall composition) may shed light into this question. 

#### Effect of Gliding Arc Plasma Jet on Cell Viability of *F. graminearum* and *F. meridionale*

Figure 5 shows confocal microscopy images used to assess cell viability of *F. meridionale* and *F. graminearum* treated for 15 min in GAPJ.

Thiazole orange cell viability indicators (green color) and propidium iodide (red color) were used, the first being responsible for penetrating the viable and non-viable cells and the second being responsible for penetrating only the non-viable cells. *F. meridionale* was more susceptible to treatment with GAPJ, as seen in Figure 4a,c,e,g. In the analysis using cell viability dyes, hyphal death was observed, as demonstrated by the total staining with PI after the 15-min treatment (Figure 5e).

The microscopy image for evaluating the cell viability of *F. meridionale* for both dyes was not added (TO+PI). TO marks both viable and non-viable cells, while PI marks only non-viable cells; as *F. meridionale* hyphae was non-viable after the 15 min GAPJ treatment, no contrast between the two dyes were visualized.

*F. graminearum* was shown to be less susceptible to treatment with GAPJ, exhibiting lower fluorescence when using the PI dye (Figure 5f) compared to the fluorescence of the TO dye (Figure 5d). Figure 5g shows the fluorescence when using the two dyes simultaneously, and the treatment with GAPJ for 15 min is not enough to kill the hyphae of *F. graminearum*.

Although the mode of action of NTAPP still requires further investigation, recent work associates the reactive species generated in the interaction between plasma and fungi with changes in the surface of hyphae and spore structure, which reduces the ability of fungi to grow [71,72].

Guo et al., 2022, showed a remarkable inhibition of *F. graminearum* viability due to RONS generated in plasma-activated water. Sculpture of the cell wall induced by oxidative stress with alteration of membrane permeability was also observed. At the cellular level, the cell wall, membrane, and mitochondria were the most affected organelles by RONS [64]. Therefore, recent studies show that NTAPP partially inhibits the viability of *F. graminearum*, which corroborates with our results.

Masiello et al. (2021) [73] used NTAPP-generated coating and NTAPP-generated coating incorporated with prothioconazole to inhibit *F. graminearum* and *F. proliferatum* infection in maize. The authors showed that an NTAPP-generated coating with incorporated prothioconazole was the best approach for protecting corn seedlings against these fungi [73]. Therefore, depending on the food matrix and the *Fusarium* species associated, there is a need to join more than one technology to control the colonization.

### 3.2. Effect of Gliding Arc Plasma Jet on the Mycobiota of Barley Grains

The mycobiota of barley grain samples submitted to the treatment by GAPJ for 10, 20, and 30 min under 3000 rpm of continuous rotation (Figure 2) was evaluated. It is important to note that the exposure times to GAPJ were based on the results obtained for radial growth and cell viability.

Table 2 shows the results of untreated and GAPJ-treated samples. High fungal contamination was observed in the untreated samples: in descending order, yeasts, FGSC, *Alternaria* spp., and *Aspergillus* spp. After 10 min of treatment with GAPJ, a reduction of 1 Log CFU/g was observed for *Alternaria* (*p* < 0.05). For yeasts, a slight count reduction was observed, but without statistical difference (*p* > 0.05). A reduction of approximately 1 Log CFU/g for yeasts and the FGSC was observed after 20 min of GAPJ exposure (*p* < 0.05). The exception was *Aspergillus* spp., which had shown low contamination before exposure to GAPJ. With increasing exposure time to GAPJ, precisely in 30 min, a reduction of 2 Log CFU/g was observed for yeasts and *Alternaria* (*p* < 0.05). Even though the 30-min treatment showed lower fungal counts, no statistical difference was observed between the 20- and 30-min treatments for all the filamentous fungi recovered from barley samples.

The FGSC was the second most frequent filamentous fungi found in barley grains not treated with GAPJ. The reduction of 1 Log CFU/g after the 20 min of GAPJ exposure demonstrated that this technology could possibly aid the controlling strategies for this fungus.

Our results have shown that although the most significant reduction in yeast contamination was after the 30 min treatment, no significant differences were observed between 20 and 30 min of GAPJ exposure for filamentous fungi. Therefore, under the conditions tested, the exposure time to GAPJ of 20 min can be considered an adequate time for controlling filamentous fungi, including the FGSC.

NTAPP as an antimicrobial agent in grains and seeds has been applied for approximately two decades and has shown an antimicrobial effect. It is worth noting that a variety of plasma reactors are used for these applications. As an example, Zahoranová et al., 2018, observed reduced contamination by bacteria and fungi (*Aspergillus flavus*, *Alternaria alternata*, and *Fusarium culmorum*) in corn after treatment with non-thermal ambient pressure plasma (NTAPP) generated by discharge coplanar diffuse surface barrier (DCSBD) [74]. Likewise, Kordas et al. (2015) and Selcuk et al. (2008) reported a significant inhibition of fungal contamination in wheat grains using NTAPP [75].

Recently, Guo et al., 2022, showed that plasma-activated water, which is less efficient than direct exposure to plasma, was effective for controlling *F. graminearum*. The authors proposed a potential antifungal mechanism from the point of view of cellular response and differential gene expression. According to the authors, RONS compromise the cell membrane, leading to the intracellular accumulation of reactive oxygen species (ROS) and subsequent reduction of intracellular pH, leading to depolarization and mitochondrial dysfunction [64]. DESeq2 sequencing analysis reinforced this hypothesis [64].

In the current study, we used the GAPJ and compressed air as working gas to treat naturally contaminated barley grains; in addition to the long-lived, medium-life, and short-lived RONS, the influence of UV radiation also occurs, which helped the antifungal activity observed [2]. As previously mentioned, reactive species play an essential plasma-surface interaction, promoting antimicrobial effects. In the case of seeds/grains, there is a need for further studies of the food matrix effect on the survival of filamentous fungi and yeast spores, as well as on the seed/grain surface properties (i.e., resistance and hardness); thus, cold plasma can be efficiently used as a chemical-free method for controlling fungal contamination.

### 3.3. Effect of Gliding Arc Plasma Jet on Deoxynivalenol and Deoxynivalenol-3-Glucoside

Table 3 shows the DON and D3G contamination profile in 20 samples of naturally contaminated barley grains and treated with GAPJ for 20 min. Twenty barley samples with detectable levels of DON were selected [51]; nevertheless, not all of the samples were contaminated by D3G (Table 3). Based on the previous analyses, the 20 min GAPJ treatment was selected.

Overall, the results showed a reduction in DON contamination between 1% and 89%; in 14 samples (70%), the reduction was above 50%. Nevertheless, in four samples, an increase between 2% and 33% of DON levels was observed (Table 3).

The reduction of DON levels in most samples, as shown in Table 3, indicates that GAPJ is a potential strategy to mitigate DON in grains. However, further studies are necessary to determine the compounds formed after the GAPJ treatment of barley grains contaminated with DON. A significant increase (*p* < 0.05) in the concentration of D3G might indicate that DON was converted into this compound after GAPJ treatment.

It is important to note that D3G can be reconverted to DON by the intestinal microflora of mammals, which can lead to an increase of the total exposure to DON [76]. Previous studies have demonstrated DON and D3G contamination in barley [77,78], with a correlation between DON reduction and D3G formation in host plants and during food processing [77]. In contrast, a single study demonstrated that barley samples contaminated by DON and D3G experienced a significant reduction in the levels of both mycotoxins after NTAPP treatment [79].

Kříž et al., 2015, effectively reduced the concentration of DON and D3G in malting barley seeds using two types of plasma: (i) plasma generated by microwaves at low pressure, and (ii) gliding arc in atmospheric pressure [80]. Barley grains were exposed to the treatment of both plasmas for 4 min, which resulted in about 80% and 50% reductions in DON concentrations, respectively. Treatment was also effective for D3G and T2 toxins, but the rate of toxin deactivation depended on the type of toxin and the plasma generating reactor.

## 4. Conclusions

Based on the results found in this study, GAPJ demonstrated potential for inactivating *F. meridionale* and *F. graminearum* for 15 min of exposure. Cell viability tests showed that the 15-min GAPJ treatment completely inactivated *F. meridionale,* whereas *F. graminearum* was resistant. Barley grains naturally contaminated by fungi and mycotoxins treated for 20 min with GAPJ showed a reduction of up to 2 log CFU/g for yeasts, *Alternaria*, and the FGSC. In addition, there was a reduction in DON levels of up to 89%, which can be considered a strategy for controlling DON in grains. However, an increase of D3G levels in the samples was observed, indicating potential conversion from DON to D3G. More studies will be needed to identify the modified mycotoxins formed after GAPJ treatment in barley, as well as other food matrices, to determine the use of this technology for mycotoxin mitigation in the food industry.

## Figures and Tables

**Figure 1 ijerph-20-05072-f001:**
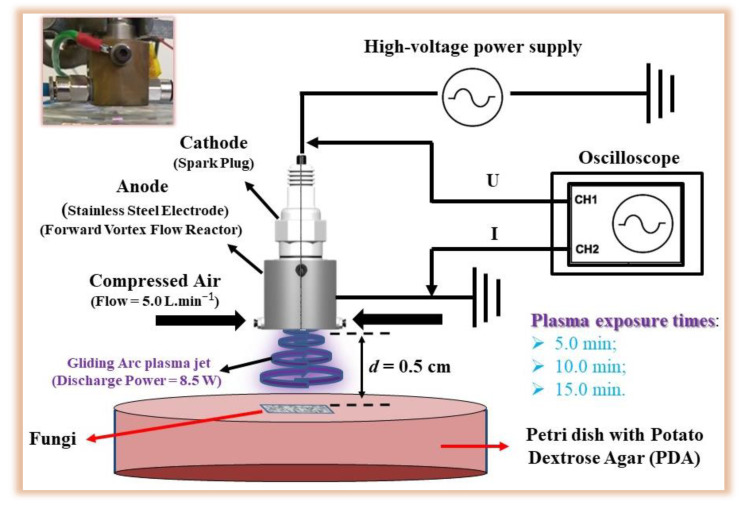
Gliding arc plasma jet experimental setup for treatment of *F. graminearum* (FML08) and *F. meridionale* (FML03). A photo depicting the setup of the plasma jet used for treatment was added to the figure.

**Figure 2 ijerph-20-05072-f002:**
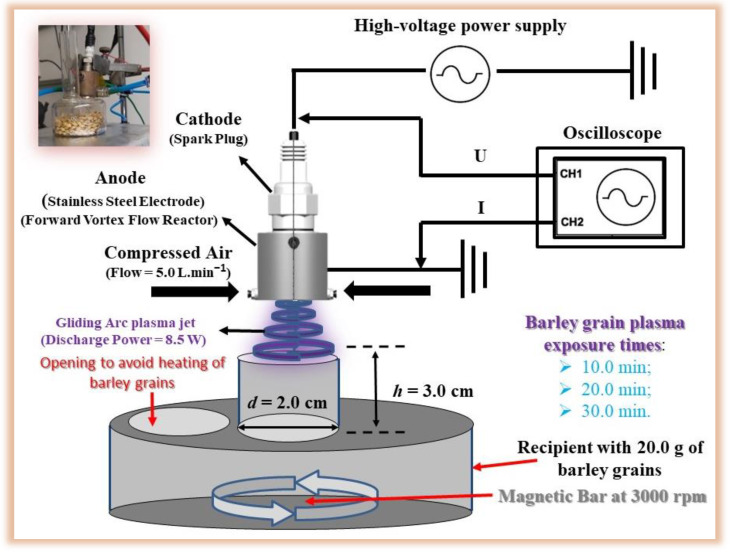
Gliding arc plasma jet experimental setup for treatment of barley grains at 3000 rpm with the aid of a magnetic stirrer device. A photo depicting the setup of the plasma jet used for treatment was added to the figure.

**Figure 3 ijerph-20-05072-f003:**
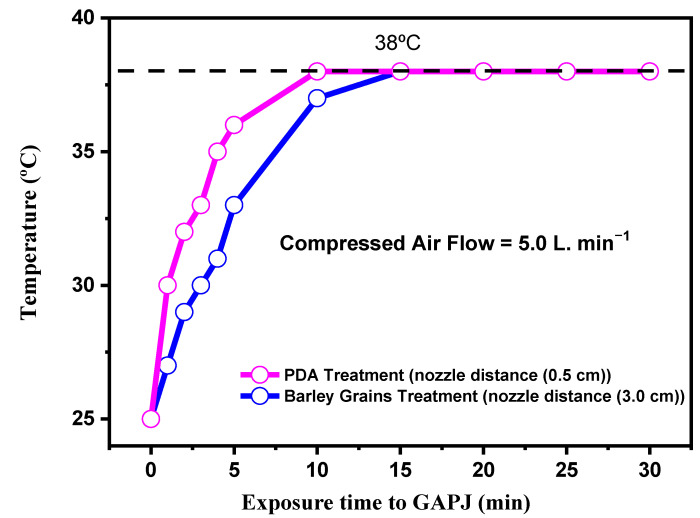
Temperature as a function of exposure time to GAPJ for both PDA and barley grain treatments.

**Figure 4 ijerph-20-05072-f004:**
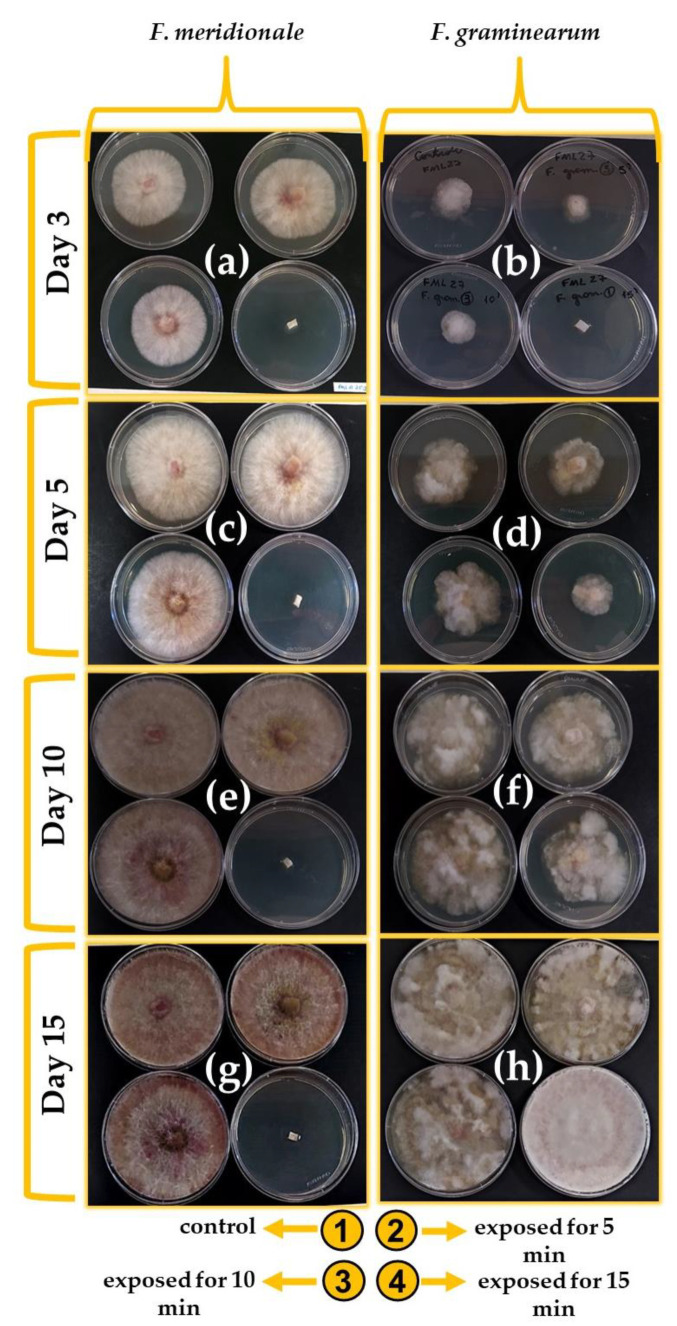
Monitoring the radial growth of the fungi *F. meridionale* (FML03) and *F. graminearum* (FML08) at 25 °C for 3, 5, 10, and 15 days. (**a**,**c**,**e**,**g**) *F. meridionale* treated by GAPJ for 5, 10, and 15 min. (**b**,**d**,**f**,**h**) *F. graminearum* treated by GAPJ for 5, 10, and 15 min.

**Figure 5 ijerph-20-05072-f005:**
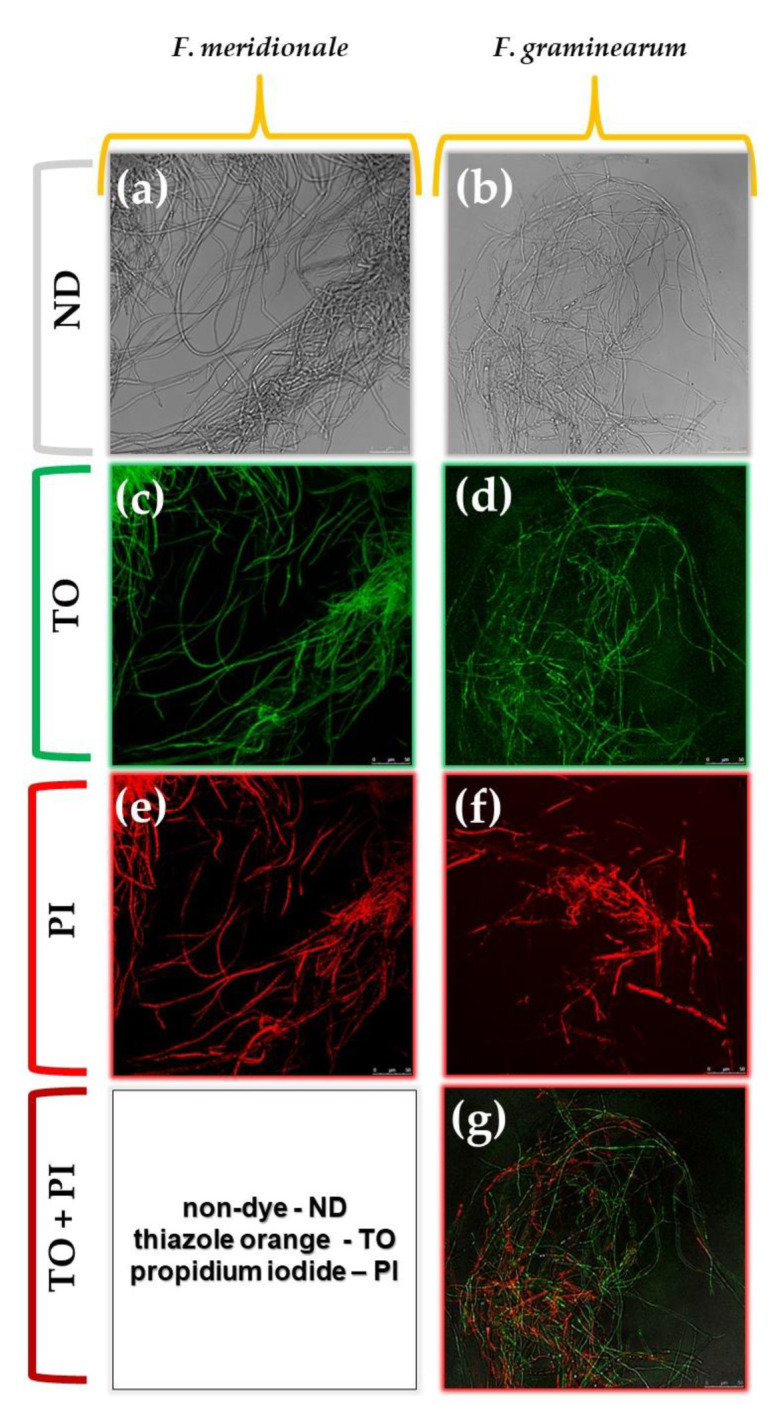
Confocal microscopy images of *F. meridionale* (FML03) and *F. graminearum* (FML08) treated by GAPJ for 15 min (40× magnitude): (**a**,**b**) without dye; (**c**,**d**) with thiazole orange (green color); (**e**,**f**) with propidium iodide (red color); and (**g**) with TO + PI.

**Table 1 ijerph-20-05072-t001:** Gliding arc plasma jet (GAPJ) parameters.

GAPJ Parameters	Values
discharge power (W)	8.5
frequency (kHz)	19.2
peak-to-peak current (mA)	2.4
peak-to-peak voltage (kV)	3.5
compressed air flow (L/min)	5
nozzle distance (cm)	0.5

**Table 2 ijerph-20-05072-t002:** Mean Log CFU/g (five samples/treatment) of fungi isolated from barley grains not treated and treated during 10, 20, and 30 min with GAPJ.

Time of Exposure to GAPJ	Yeasts (Log UFC/g)	*Alternaria* spp. (Log UFC/g)	FGSC ** (Log UFC/g)	*Aspergillus* spp. (Log UFC/g)
0 min *	5.90 ^d^ ***	3.38 ^g^	2.85 ^j^	1.08 ^l^
10 min	5.61 ^d^	2.32 ^h^	2.85 ^j^	0.54 ^m^
20 min	4.73 ^e^	1.88 ^i^	1.67 ^k^	0.62 ^m^
30 min	3.26 ^f^	1.53 ^i^	1.75 ^k^	0.52 ^m^

* Sample not treated by GAPJ. ** FGSC: *Fusarium graminearum* species complex. *** Equal letters mean no statistical difference between treatments, and different letters indicate the difference between treatments by Tukey’s test (*p* < 0.05).

**Table 3 ijerph-20-05072-t003:** Levels of deoxynivalenol (DON) and deoxynivalenol-3 glycoside (D3G) in naturally contaminated barley samples and after GAPJ treatment for 20 min.

Samples	Samples Not Treated by GAPJ		Samples Treated by GAPJ for 20 min		∆(DON (%)
	DON (µg/Kg)	D3G (µg/Kg)	DON (µg/Kg)	D3G (µg/Kg)	
1	828.4	35	258	930	−69
2	554.8	20	600	1050	+8
3	261.1	15	332	800	+27
4	1293.6	15	373	460	−71
5	1836.2	<LOD *	409	1090	−78
6	2131.1	<LOD	404	1090	−81
7	897.8	<LOD	400	2450	−55
8	1174.6	LOD	450	570	−62
9	873.2	15	292	510	−67
10	2074.4	<LOD	230	1980	−89
11	1732.7	<LOD	460	930	−73
12	1572	39	380	840	−76
13	880.3	22	870	420	−1
14	674.5	21	690	400	+2
15	1062.9	32.1	520	300	−51
16	1945.2	<LOD	660	1110	−66
17	1897	84.1	464	300	−76
18	554	15	740	350	+34
19	871.1	15	595	810	−32
20	1240.6	<LOD	222	850	−82

* Limit of detection (LOD = 15 µg/Kg).

## Data Availability

The data that support the findings of this study are available upon reasonable request.

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
