# Peer review of "Effect of Gliding Arc Plasma Jet on the Mycobiota and Deoxynivalenol Levels in Naturally Contaminated Barley Grains"

_ijerph, 2023, doi:10.3390/ijerph20065072_

Round 1

Reviewer 1 Report

MANUSCRIPT ID:  ijerph-2041428

Title: Effect of gliding arc plasma jet on the mycobiota and deoxynivalenol levels in naturally contaminated barley grains.

Submitted for publication to: International Journal of Environmental Research and Public Health

In the present work the effect of the cold plasma treatment on the pathogen cause of DON contamination and the DON concentration is studied using several exposed times in two locally isolated toxigenic strains of Fusarium species and in naturally contaminated barley grains. The efficacy of the treatment is evaluated measuring parameters that are related to the pathogen itself (radial growth, cell viability, fungi survival) as well as the mycotoxin production, quantitively.

The research plan is very well described and clearly demonstrates a total approach of the topic. The obtained results are objectively discussed and are documented using recent literature data. Moreover, the manuscript is professionally written and the obtained conclusions merit particular utility. However, some questions are raised and need to be clarified in order the obtained results and conclusions be more explicit and justified.

Minor concerns:

Materials & Methods

2.1.: What is the incubation temperature for the growth of FML03 and FML08 on PDA medium? It could be mentioned.

2.2 How were the naturally contaminated barley grain samples purchased? Was there any other preparation of the collected samples prior to storage? How were the samples were transferred? What does RS-Brazil mean (an explanation for the abbreviation is not given)? Some details could be provided.

Results

3.1.1.: A further explanation for not showing a microscopy image for the assessment of the cell viability of F. meridionale when both dyes were added (TO+PI) could be given. It’s not clear.  

Additional notices concerning the text format:

  1. Wrong numbering of the Tables displayed either in the text reference (Line 354: Table 2 instead of Table 3) or in the titles of the Tables.
  2. The scientific names of the fungal species are not italicized everywhere in the text e.g. Line 128 and Line 262.

Reviewer 2 Report

The experimental method was detailed described in the manuscript. To the reviewer's knowledge, the results look reasonable. The work showed that a gliding-arc plasma jet could decontaminate fungi in barley, which agreed with some previous related investigations. However, several questions need to be addressed.

(1) Photo of the plasma treatment should be displayed in Fig.1 and Fig. 2.

(2) RONS, such as Ozone and NOx could create negative effect on barley grains for 10 to 30 mins, so please show the necessary indicators.

(3) It should be clearly described whether the gliding-arc plasma has directly contacted with samples on the PDA in Fig. 1.

(4) Data on temperature rise during treatments should be presented. The authors need to clarify whether heat or plasma was the deciding factor for sterilization.

(5) The manuscript mentioned (line 248 to line 251’ the optical emission spectrum of the GAPJ 248 contains NH, N2, and OH species in abundance.”) emission spectrum of gliding-arc plasma, however, commonly, NOx should be the major component of RONS. Ozone should be in second place. So more accurate discussion is required.

Reviewer 3 Report

Comments:

Comments:

- justify why these two species were selected for the study, especially F. meridionale, and what is its harmfulness to barley cultivation (in soybeans it is about 12% and in barley?)

- expand in the introduction about the importance of these species for barley cultivation, emphasize their importance (e.g. yield loss caused by them)
- line 230-232: no visible effect after 5 and 10 minutes. (based on Figure 3), it is visible only after 15 min. Please attach data in the table, where the significance of these differences will be marked, confirming these conclusions

- line 245-247: what components of the wall differ between the two species, that we have differences in susceptibility,

- line 361-362: if so, why is it not evident in all samples naturally contaminated with D3G?

- line 388-389: in Results, you also write about such action after 5 and 10 minutes (please specified),

- line 389-390: only on the basis of Fig. 4e?, I would be inclined to agree if there was a diagram for F. meridionale TO+PI that would confirm it (as is Fig. 4g for F. graminearum).

Round 2

Reviewer 3 Report

The authors gave exhaustive answers to all my comments and made the necessary corrections in the text. The work is interesting and I wish it had many people interested in this research.